# Role of the Glycosylphosphatidylinositol-Anchored Protein TEX101 and Its Related Molecules in Spermatogenesis

**DOI:** 10.3390/ijms21186628

**Published:** 2020-09-10

**Authors:** Hiroshi Yoshitake, Yoshihiko Araki

**Affiliations:** 1Institute for Environmental & Gender-specific Medicine, Juntendo University Graduate School of Medicine, Urayasu, Chiba 279-0021, Japan; hyositak@juntendo.ac.jp; 2Department of Obstetrics & Gynecology, Juntendo University Graduate School of Medicine, Bunkyo, Tokyo 113-8421, Japan; 3Division of Microbiology, Department of Pathology and Microbiology, Nihon University School of Medicine, Itabashi, Tokyo 173-8610, Japan

**Keywords:** glycosylphosphatidylinositol-anchored protein, TEX101, fertilization, testicular germ cell

## Abstract

Glycosylphosphatidylinositol (GPI)-anchored proteins (APs) on the plasma membrane are involved in several cellular processes, including sperm functions. Thus far, several GPI-APs have been identified in the testicular germ cells, and there is increasing evidence of their biological significance during fertilization. Among GPI-APs identified in the testis, this review focuses on TEX101, a germ cell-specific GPI-AP that belongs to the lymphocyte antigen 6/urokinase-type plasminogen activator receptor superfamily. This molecule was originally identified as a glycoprotein that contained the antigen epitope for a specific monoclonal antibody; it was produced by immunizing female mice with an allogenic testicular homogenate. This review mainly describes the current understanding of the biochemical, morphological, and physiological characteristics of TEX101. Furthermore, future avenues for the investigation of testicular GPI-Aps, including their potential role as regulators of ion channels, are discussed.

## 1. Introduction

Mammalian spermatozoa acquire fertilizing capacity as they cross the epididymis in vivo [1,2,3]. However, spermatogenesis itself takes place in the testicular seminiferous tubules [4]. Spermatogenesis, i.e., testicular mitotic proliferation, meiosis, sperm cell division, and morphological changes from haploid sperm to mature spermatozoa, results in the generation of cells that are highly specialized in structure and function [4]. No other cell has undergone such extreme morphological changes while undergoing both genetic modification and reduced chromosomal pluripotency. It is generally believed that the testes have a unique mechanism to control spermatogenesis, where certain molecules are activated during germ cell production and others are repressed.

There is increasing evidence that glycosylphosphatidylinositol (GPI)-anchored proteins (APs) on the plasma membrane are involved in the function of a range of cell types [5], including gametes. For example, male infertility is caused by disruption of the testicular angiotensin-converting enzyme, which releases GPI-APs [6]. Thus, GPI-APs in the testis are quite important for proper sperm–plasma membrane structure formation, which influences normal fertilizing ability.

In the past two decades, we have studied the structure and function of testicular factors, especially the mechanism of germ cell production, starting with the discovery of a protein (TEX101). This protein is the core of our research, and we have accumulated a wealth of knowledge based on the functional relationship between TEX101, a GPI-AP, and other molecules.

In this review, we focus on TEX101, which belongs to the lymphocyte antigen 6 (Ly6)/urokinase-type plasminogen activator receptor (uPAR)-(LU) protein superfamily [7,8]; specifically, we discuss this as an important biomolecule for the production of normally functioning gametes. TEX101 was originally identified as a glycoprotein that contained the antigen epitope for a specific monoclonal antibody (mAb); it was produced by immunizing female mice with an allogenic testicular homogenate [9,10]. Using immunoprecipitation (IP)–liquid chromatography (LC)–tandem mass spectrometry (MS/MS) and immunofluorescence studies, we confirmed the association of TEX101 with at least two proteins in the plasma membranes of adult testicular seminiferous tubules: cellubrevin (a member of soluble *N*-ethylmaleimide-sensitive factor attachment protein receptor family) and Ly6k (a GPI-AP that was recently recognized as an LU protein) [11,12]. TEX101 is a unique germ cell marker that is expressed only during gametogenesis; its exhibits sexually dimorphic expression in developing gonadal tissues [13]. Before focusing on TEX101, we provide a brief explanation concerning the molecular nature of GPI-AP for understanding the scientific status of a class in membrane proteins.

## 2. GPI-APs

### 2.1. History of the Discovery

In 1960, Slein and Logan reported that the intravenous injection of purified components of *Bacillus anthracis* toxin into rabbits increased levels of alkaline phosphatase in their serum [14]. Moreover, phospholipase C (PLC) from *Bacillus cereus* culture medium induced the phosphatemia in those rabbits [15]. In 1976, Ikezawa and his colleagues purified phosphatidylinositol (PI)-specific PLC from *B. cereus* culture supernatant and found that the enzyme released alkaline phosphatase from plasma membranes of rat kidney cells [16]. Their research group also showed elevated alkaline phosphatase levels in rat serum following the intravenous injection of PI-PLC [17]. These results suggested that PI-PLC treatment removes alkaline phosphatase from cell plasma membranes. Subsequently, PI-PLC was found to induce the release of acetylcholinesterase, 5′-nucleotidase, and alkaline phosphodiesterase I from mammalian cell plasma membranes [18,19,20], indicating that these proteins are bound to the plasma membrane with phosphatidylinositol. Ferguson et al. discovered that variant surface glycoproteins of *Trypanosoma brucei* are anchored to the plasma membrane through C-terminus linkage with glycosyl-1,2-dimyristyl PI, which contains mannose, glucosamine, ethanolamine, and PI [21]. Several subsequent studies provided evidence that GPI-APs form a class of similar membrane proteins.

### 2.2. Basic Structure of GPI-APs

The plasma membrane contains many types of proteins, which are involved in cell functions and interactions with the cell environment. Membrane proteins are associated with the cell membrane lipid bilayer in various ways [22]. Many proteins on the cell surface exist as transmembrane proteins, which contain both intracellular and extracellular domains. However, some membrane proteins lack intracellular domains; they may be tethered to the outer leaflet of the plasma membrane by GPI, and these are called GPI-APs. The core structure of the GPI anchor consists of PI, glycans (glucosamine and three mannose residues), and phosphoethanolamine (Figure 1). Proteins bind to GPIs through an amino bond between the peptide C-terminus and the ethanolamine amino group (Figure 1).

### 2.3. General Characteristics and Potential Biological Significances of GPI-APs

GPI-APs are broadly conserved among eukariotes [23], comprising approximately 170 types of human and mouse proteins (UniProt, http://www.uniprot.org). GPI-APs are presumed to play pivotal roles in fertilization [6,24,25,26], the immune system [27,28,29], nerve formation [30,31,32], and embryonic development [33,34,35], where they function as enzymes (e.g., alkaline phosphatase, 5′-nucleotidase (CD73), and dipeptidase (DPEP)) [36], cell adhesion molecules (e.g., lymphocyte function-associated antigen 3 (CD58) and neural cell adhesion molecules) [37,38], component regulatory proteins (e.g., CD55 and CD59) [39,40], receptors (e.g., CD14, CD16b, and folate receptor) [41,42,43], mammalian antigens (e.g., Thy-1 (CD90), carcinoembryonic antigen (CD66e) [23,44], and ion channel or its modulator (e.g., α2δ subunit of voltage-gated calcium channel, prion protein, and Lynx1) [45,46,47].

Although GPI-APs lack an intracellular domain, a few hypotheses have been proposed to explain the mechanism by which they transduce extracellular signals into the cytoplasm. One tentative theory is that outside-in signaling mediated by GPI-APs passes through the plasma membrane by means of an associated transmembrane molecule. For example, uPAR (CD87), lipopolysaccharide (LPS)/LPS binding protein (LBP) receptor (CD14), Fcγ receptor IIIB (CD16b), and GPI-80 are associated with β_2_ integrin (CD11b/CD18) on the surfaces of polymorphonuclear leukocytes; these interactions regulate integrin-mediated cell adhesion [48].

A more likely theory is that lipid rafts are involved in signal transduction regulated by GPI-APs. Lipid rafts are membrane domains that contain many GPI-APs, Src family kinases, transmembrane proteins, cholesterol, and sphingolipids [49]; lipid rafts are presumed to play roles in signal transduction, cell adhesion, migration, and protein trafficking [50,51,52]. GPI-APs frequently cluster in lipid rafts and transiently recruit Lyn (a Src family member associated with lipid rafts) by means of both protein–protein and lipid–lipid (raft) interactions [53]. Indeed, GPI-AP clustering is known to induce cell activation via tyrosine phosphorylation [54,55,56,57]. Most GPI-APs are post-translationally modified by the addition of oligosaccharide (OS) chain(s) (UniProt, http://www.uniprot.org). Recently, Miyagawa-Yamaguchi et al. reported that GPI-APs with high-mannose or complex-type *N*-linked OS chains cluster on distinct lipid rafts under different physiological conditions [58]. This finding suggests that different GPI-AP types may form individual lipid rafts depending on the *N*-glycosylation pattern.

Many GPI-APs (e.g., CD14, CD16b, CD55, uPAR, and GPI-80) have been reported to exist on cell surfaces as membrane proteins and in extracellular fluid in soluble form [59,60,61,62,63]. Although some soluble GPI-APs are useful as biomarkers of cancers or inflammation diseases [64,65,66,67,68,69,70,71], the precise functions of soluble GPI-APs remain unclear. Further investigations are needed to understand the biological functions of these molecules.

### 2.4. GPI-APs in the Testis

Thus far, the UniProt database (http://www.uniprot.org) lists 29 GPI-APs that are expressed in the human and mouse testes (Table 1). Among these molecules, DPEP3, a glioma pathogenesis-related protein 1 (GLIPR1)-like protein 1, hyaluronidase PH-20, hyaluronidase-5, Ly6k, prion-like protein doppel, serine protease 41, sperm acrosome membrane-associated protein 4 (SPACA4), TEX101, and testisin are strongly expressed in the male gonad (i.e., testis, epididymis, or mature spermatozoa), but they are exhibited weakly or negligible in other tissues (e.g., female gonads and somatic organs) [9,25,72,73,74,75,76,77,78,79]. Among these, GLIPR1-like protein 1, PH-20, and SPACA4 are presumed to participate in the interaction between sperm and oocyte [73,79,80]. Although the other GPI-APs listed may be involved in male germ cell development and differentiation, the precise functions of most of these molecules remain unclear. As a hypothesis, the LU protein superfamily such as Ly6k and TEX101, which possesses three-fingered protein domain (TFPD) [81], may have a function as a regulator of ion channels. Indeed, Lynx1, a GPI-AP co-localized with α7 and α4β2-nicotinic acetylcholine receptors (nAChRs), is the first identified prototoxin having TFPD in the brain, which modulates the function of nAChRs [47]. In the testes, it has been well documented that ion channels, such as CatSpers, play an important role in sperm functions [82,83,84]. In fact, various ions are mediated with important functions of sperm, such as the acrosomal reaction and hyperactivation of the motility, so that ion channels are deeply involved in control. With respect to TEX101, recent crystal structure analysis provides direct evidence that this molecule actually has two LU domains, both of which have a TFPD [85]. It is striking that such structural analysis holds great promise for elucidating the actual interactions between this molecule and a group of molecules associated with the ion channels. However, the main points remain unclear, and the physiological functions of these molecules are far from completely understood. Accordingly, GPI-APs in the testis as these modulators could be a focus for further research, including studies of ion channels.

## 3. Significance of TEX101 in the Fertilization Process

### 3.1. Strategies for Identifying Testis-Specific Molecule(s)

Generally, researchers consider tissue-specific factors to be localized in tissues related to their specialized functions. Since the gonad is the sole organ of germ cell production, important factors for spermatogenesis are expected to be located in testicular germ cells (TGCs). In a previous study of the molecular mechanisms concerning gametogenesis and the fertilization process, we used mouse testes as an experimental material to identify such factors, because adult seminiferous tubules contain germ cells at all stages. To find out specific factor(s), we did not choose a genetic approach, because we know from our professional experience that the establishment of a specific molecular probe (such as Abs) is a key step for the further characterization of novel molecules. As producing mAbs with satisfactory performance for further molecular characterization was not always easy, we did not dare use a strategy that used a genetic approach. Using splenocytes from female mice immunized with syngenic male germ cells, we established several mAbs and first performed immunohistochemical staining (IHS) of the testis. Against the mAbs established, we did further Western blot (WB) and IP analyses, which was followed by micro-amino acid analysis (this process has now been effectively replaced by MS analysis) to find out unique factor(s). This procedure can be expected to find “novel” molecule(s) or molecular complexes, and the mAbs established can be expected to have almost perfect performances (compatible for IHS, WB, and IP) for further biochemical as well as morphological analyses.

We detected several hybrydoma clones that corresponded to an interesting IHS pattern within the testis. Among the hybridoma clones established, a protein detected by an mAb (IgG1, termed TES101) was produced from a hybridoma; this protein was tentatively named as TES101-reactive protein (RP); later, the nomenclature was changed to TEX101 [9,13].

### 3.2. Molecular Characteristics of TEX101, a Unique Glycoprotein Germ Cell Marker

TEX101 is composed of a signal peptide region of 25 amino acids and a mature protein region of 225 amino acids, thus forming a protein with a molecular weight of 24,093 in mice. TEX101 contains four potential *N*-glycosylation sites (Asn-Xaa-Ser/Thr) as well as many (>40) Ser/Thr residues, which could be possible *O*-glycosylation sites [9]. When we first described this molecule in 2001, the cDNA sequence revealed no homologous molecules in the DNA database; the molecule has since been classified as a member of the LU protein superfamily [7,8], based on the conserved position of cysteine residues within the molecule. The position of cysteine residues is highly conserved among major mammalian species including human, indicating similarity in the steric structure of these orthologues due to disulfide bonds (Table 2).

TEX101 contains strong hydrophobic portions [104] at both the N- and C-terminal ends of the molecule, which is typical of GPI-APs [105]. Morphological analysis of testicular tissues and molecular biological analysis of TEX101-expressing transfectants revealed that the enzyme PI-PLC, which removes surface GPI-APs, exhibited TEX101-releasing activity on the cell surface [10]. Although TEX101 is primarily detected on the TGC cell membrane [9,13], TEX101 appears to have at least two different forms in nature: GPI-AP and non-membrane-bound soluble forms—subcellular TEX101 was found in the Triton X-100-soluble fraction from the testicular membrane, as well as in water-soluble and extracellular fractions [9].

TEX101 has been known to be a glycoprotein since its identification [9,10] as described above. Various glycoproteins are generally presumed to be involved in mammalian physiological processes, from fertilization to implantation [106,107,108,109]. Glycosylation is among the most pivotal post-translational modifications; it is involved in biological processes including cell–cell interactions, as well as cell differentiation and proliferation [110].

During antigen-epitope analyses of an anti-sperm auto-Ab using spleen cells of aged mice (over one year) maintained under conventional conditions, we unexpectedly established a mAb, termed Ts4, which reacts with the OS moiety of TEX101 [111]. The Ts4 mAb serves as an auto-Ab against acrosomal regions of cauda epididymal spermatozoa [111]. At that time (two decades ago), quantitative molecular identification was quite poor; therefore, we attempted the molecular identification of a Ts4 target in testicular extract, rather than extracts of cauda epidydimal spermatozoa. In the male mouse gonad, Ts4 exhibits immunoreactivity against several types of glycoproteins in the acrosomal region of epididymal spermatozoa as well as against germ cells within seminiferous tubules by interacting with a common OS chain on the molecules [112,113]. The antigenic determinant for Ts4 was located on the fucosylated agalacto-biantennary complex-type *N*-glycan with bisecting N-acetylglucosamine (GlcNAc) of TEX101 [113]. Although TEX101 does not appear in mature epididymal spermatozoa [9,112,114] (Figure 2), we analyzed a Ts4-reactive glycoprotein in mouse cauda epididymal sperm. IP and LC-MS/MS analyses showed that alpha-*N*-acetylglucosaminidase (Naglu; a degradation enzyme of heparan sulfate) was among the glycoproteins recognized by Ts4 in epididymal spermatozoa. Using a similar strategy, we recently identified the direct target protein of Ts4 as NUP62; we also characterized GPI-AP molecular formation, including TEX101 and its related molecules, during testicular development [115].

Direct evidence of the biological significance of OS chain detected by Ts4 has not been reported. However, experimental observations increasingly suggest the potential importance of Ts4 target in the fertilization process; (1) this mAb affects fertilization in vitro [116]; (2) the molecular epitope for Ts4 showed unique structure [113]; and (3) its occurrence is limited to reproductive-related organs [112,113,115,116]. Indeed, the bisecting GlcNAc structures have been already reported to possess biological functions, such as cell growth and adhesion by modulating membrane glycoproteins [117,118,119]. Together, these findings suggest that TEX101, as the major testicular molecule possessing Ts4-reactive OS, is essential for elucidating its molecular function and mechanism.

### 3.3. Subcelllar Localization of TEX101 within Gonadal Organs

TEX101 was initially identified in the TGCs [9]; however, it is also expressed in other cells. During embryonic development, TEX101 appears in germ cells of both male and female gonads after the pregonadal period [13] (Figure 2). In the testis, TEX101 is constitutively expressed on surviving prospermatogonia during prespermatogenesis. Following the initiation of spermatogenesis, prospermatogonia differentiate into spermatogonia; TEX101 expression diminishes in spermatogonia, but it is enhanced in spermatocytes and spermatids. TEX101 is also expressed in female germ cells until the start of folliculogenesis (before birth), but it is not detected in oocytes surrounded by follicular cells within the ovary [13]. These findings imply that TEX101 exhibits sexually dimorphic expression in male and female germ cells during gonadal development. The gene name is currently registered as “*Testis Expressed 101*” in the Mouse Genome Informatics (MGI) database (The Jackson Laboratory, Bar Harbor, ME, USA), despite its expression in premature female germ cells. Thus, TEX101 should not be regarded as a specific marker for male germ cells. TEX101 (TES101RP) is a marker specific for both male and female germ cells during gonad development; however, in adult animals, it is found only in the testes [9,13].

In the testis, TEX101 has been detected in cells from seminiferous tubules, but not from interstitial tissues, including the Leydig cells [9,13]. The intensity of TEX101 immunofluorescence observed in seminiferous epithelium was nearly identical among various stages, but it is varied among cell types in seminiferous tubules. Spermatogonia attached to the basal lamina of the seminiferous tubules showed negative staining compared with more advanced cells (Stage V). However, TEX101-positive cells attached to the basal lamina were identified as leptotene or zygotene spermatocytes (Stage X). Sertoli cells were TEX101-negative at all stages [9,114].

When spermatogenesis in the testis is complete, the TEX101 protein remains on the cell surfaces of step 10–16 spermatids and testicular sperm, including the tail portion. However, TEX101 is shed from epididymal sperm in the caput epididymis [114]. In the cauda epididymis, TEX101 is no longer detectable on the male germ cells (Figure 2).

### 3.4. Molecules Associate with TEX101 in Male Germ Organ

In general, protein molecules perform their physiological actions via interactions with other proteins [120,121]. The elucidation of the functional and physical networks among proteins has fundamental importance for understanding their functions and their regulatory mechanisms [122]. Using IP followed by LC-MS/MS analyses, we identified several molecules associated directly or indirectly with TEX101 within testicular seminiferous tubules (Table 3).

Annexin A2 is a member of the annexin superfamily with Ca^2+^-dependent phospholipid-binding ability [123]. This molecule is widely expressed in many types of cells (e.g., epithelial, endothelial, trophoblast, immune, and tumor cells), and it is involved in a variety of biological functions, including membrane organization, membrane trafficking, and Ca^2+^ ion channels [123]. In the testis, Annexin A2 is found on Sertoli cells and elongated spermatids, and it is essential for the maintenance of the blood–testis barrier and the suitable release of spermatids [124].

Ly6k is a member of the LU protein superfamily, similar to TEX101. The relationship between TEX101 and Ly6k is described below.

Cellubrevin is a member of the soluble N-ethylmaleimide-sensitive factor attachment protein receptors family, which regulates membrane trafficking and fusion [125]. Our previous study using ultrahigh-resolution immunofluorescence microscopy revealed that cellubrevin plays a role in membrane trafficking of de novo TEX101 to the cell surface [11].

DPEP3 belongs to the membrane-bound DPEP family [92], which is a group of enzymes that converts leukotriene D4 to leukotriene E4 and dissolves cystinyl-bis-glycine [126]. This molecule is present only in the testis as a GPI-labeled protein [92]. Although its biological function in vivo remains to be clarified, *Dpep3*-deficient mice are fertile [127].

CD73 (the so-called ecto-5′-nucleotidase) is a GPI-anchored protein that has the enzymatic activity of the dephosphorylation of extracellular adenosine 5′-monophosphate to adenosine via the purinergic signaling pathway [128]. The expression of CD73 is broadly observed in various tissues (e.g., brain, heart, lung, liver, kidney, colon, and placenta) and some types of immune cells, such as T cells, neutrophils, monocytes, and dendritic cells [129,130]. Although CD73 and adenosine are believed to suppress the immune response in the tumor microenvironment [131], their precise function within the testis remains unclear.

### 3.5. TEX101 Function during Fertilization

Male and female mice with *TEX101* gene disruption produce spermatozoa and oocytes with normal morphology, respectively [26], suggesting that TEX101 is not essential for the morphological formation of both male and female gametes. However, the TEX101-deficient male mice were infertile [26]. Although the in vitro fertilization ability of sperm from TEX101-null mouse seems normal, sterility is mainly caused by a defect in the migrating ability of sperm into the oviduct [26], as observed in ADAM3-null mice [132]. In the testis, TEX101 expression is restricted during prespermatogenesis and spermatocyte to testicular spermatozoa [9,13,96]; therefore, TEX101 may function as a molecular chaperone in the essential fertilization process. Thus far, TEX101 has been presumed to serve as a molecular chaperone solely for ADAM3, making it similar to CALMEGIN, CALRS, PDILT, or ADAM1a [26,133,134,135,136]. However, it should be noted that a defect in TEX101 expression also reduces the expression of Ly6k (a TEX101-associated GPI-AP) in TGCs [137] (Figure 3). The experimental evidence implies that TEX101 and Ly6k both contribute to the expression of post-translational counterpart proteins on the cell membrane.

To date, although several GPI-APs have been reported in TGCs as described above, TEX101 and Ly6k are known to be essential for the production of functionally intact sperm [26,138] among these proteins. These results are based on data in which both TEX101- and Ly6k-null mice showed phenotypes nearly identical to those of gene-disrupted mouse lines, such as *Adam3*^−/−^ [132,139]. However, the precise molecular mechanisms that cause infertility among these spermatozoa remain unknown. Indeed, our polysome analysis results suggested that *Adam3* signaling occurs in adult testes; however, translation activity was not detected [137]. In fact, almost no appropriate molecular probe for ADAM3 detection at the protein level exists, particularly with respect to morphological analyses; this limits our understanding of the molecular relationship between ADAM3 and its related molecules. In addition, the physiological role of TEX101 in female gametogenesis remains unclear. Therefore, we argue that researchers should be more cautious in asserting scientific conclusions regarding the molecular significance of ADAM3-related molecules at the protein level.

## 4. Conclusions and Future Aspects

In this review, we represented current knowledge of GPI-APs in the testis, mainly with respect to TEX101. Since GPI-APs do not possess an intracellular domain, the mechanism of GPI-AP cell–cell signal transduction, including its relation to ion channels, remains controversial. The specific roles of membrane-bound GPI-APs and their soluble form also remain unknown.

The functions of various biomolecules including GPI-APs have been elucidated since the development of gene-deficient model mice in the late 20th century. It is generally believed that biomolecule function can be fully explored only through gene-disruption model analysis; however, these techniques also have many limitations.

Biomolecules have existed “in vivo” for an extremely long time before scientists discovered them. Biomolecules also evolved “in vivo” for an extremely long time before scientists discovered them; therefore, their synthesis would be expected to be meaningful. However, “Does the existence of a molecule in an organism mean that it (its existence) is functional?”. Efforts to unravel this philosophical question continue for scientists, and it is easy to imagine that scientists will spend a large amount of time in this area, even in GPI-AP studies of testes.

## Figures and Tables

**Figure 1 ijms-21-06628-f001:**
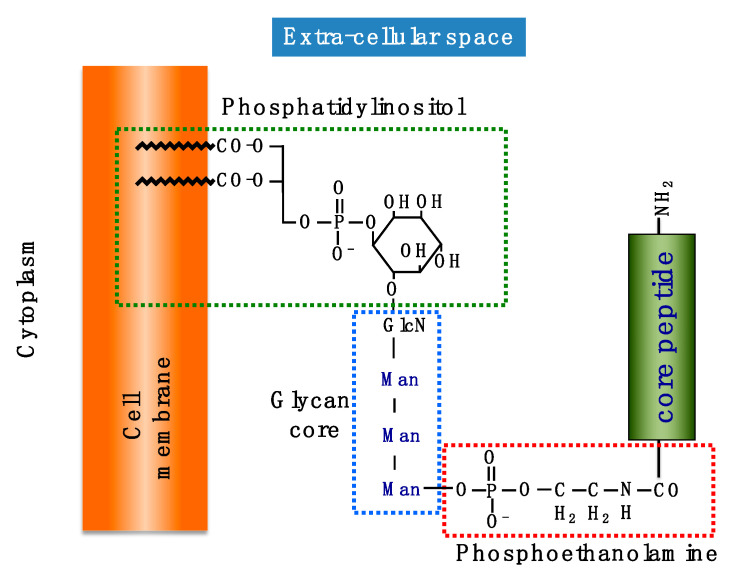
**Schema of glycosylphosphatidylinositol-anchored protein (GPI-AP) structure**. The C-terminus of core peptide binds to phosphoethanolamine followed by three mannose (Man) residues and glucosamine (GlcN). The glycan core further links to phosphatidylinositol.

**Figure 2 ijms-21-06628-f002:**
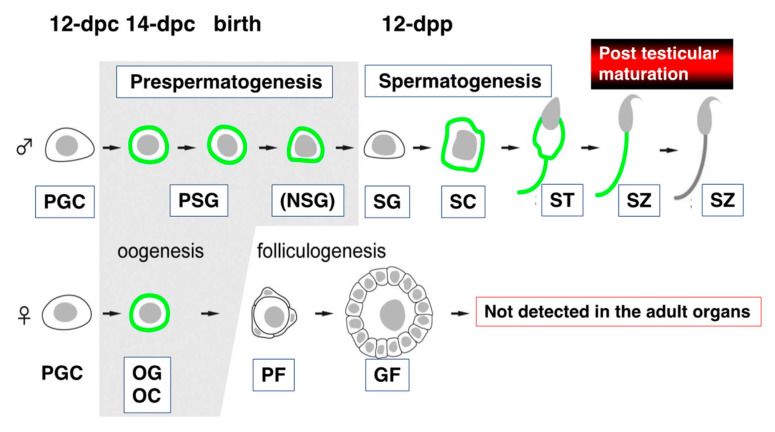
**Molecular expression of TEX101 during gametogenesis. Germ cells that express TEX101 are indicated with bold lines**. PGC, primordial germ cell; PSG, prospermatogonium or gonocyte; NSG, neonatal-type undifferentiated spermatogonium; SG, spermatogonium; SC, spermatocyte; ST, spermatid; SZ, spermatozoon, OG, oogonium; OC, oocyte; PF, primordial follicle; GF, glowing follicle (modified from [13]).

**Figure 3 ijms-21-06628-f003:**
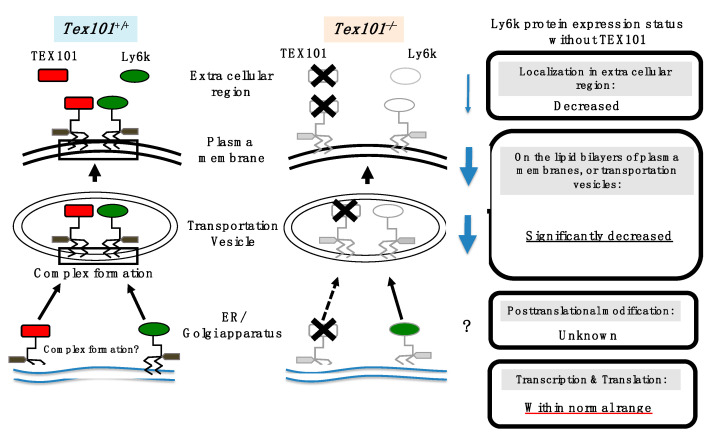
**Molecular status of Ly6k with/without TEX101 in the testicular germ cells (TGCs)**. The left diagram indicates TEX101/Ly6k complex formation of wild-type (Tex101^+/+^) mouse. After translation, GPI remodeling of these molecules is completed from endoplasmic reticulum (ER) to Golgi apparatus; then, these molecules are expressed as a TEX101/Ly6k complex (represented by black square) on lipid bilayers including a transportation vesicle and plasma membrane. In addition, (a part of) both TEX101 and Ly6k are released into extracellular space. In TEX101-null TGCs (the right diagram), Ly6k expression is drastically decreased. Black cross marks indicate the disruption of the molecules. The potential status of Ly6k protein expression without TEX101 is boxed (originated from [137]).

**Table 1 ijms-21-06628-t001:** GPI-APs expressed within the testis.

Protein Name	Putative Function	Reference
5′-nucleotidase (CD73)	Enzyme	[86]
Acetylcholinesterase	Enzyme	[87]
Alkaline phosphatase, placental-like	Enzyme	[88]
Alkaline phosphatase, tissue-nonspecific isozyme	Enzyme	[89]
CD109	Receptor	[90]
CD59	Complement regulator	[91]
Complement decay-accelerating factor (CD55)	Complement regulator	[91]
Dipeptidase 2 (DPEP2)	Enzyme	[92]
**Dipeptidase 3 (DPEP3)**	Enzyme	[72]
GDNF family receptor alpha-1	Receptor	[93]
GDNF family receptor alpha-2	Receptor	[93]
Glypican-5	Receptor	[94]
**GLIPR1-like protein 1**	Others	[73]
**Hyaluronidase PH-20**	Enzyme	[75]
**Hyaluronidase-5**	Enzyme	[75]
Lipoprotein lipase	Enzyme	[95]
Lymphocyte antigen 6A-2/6E-1 (Ly6A/E)	Others	[96]
Lymphocyte antigen 6E (Ly6E)	Receptor	[97]
**Lymphocyte antigen 6K (Ly6K)**	Others	[12]
Ly6/PLAUR domain-containing protein 6 *	Receptor	[98]
Mesothelin *	Others	[99]
**Prion-like protein doppel**	Receptor	[77]
RGM domain family member B	Receptor	[100]
**Serine protease 41**	Enzyme	[101]
**Sperm acrosome membrane-associated protein 4**	Enzyme	[79]
**Testis-expressed protein 101 (TEX101)**	Others	[9]
**Testisin**	Enzyme	[25]
Thy-1 membrane glycoprotein	Others	[102]
TNF receptor superfamily membrane 10C	Receptor	[103]

Bold type: highly specific expression in the gonad. *: Data from gene expression analysis only.

**Table 2 ijms-21-06628-t002:** The primary structure of TEX101 in major mammalian species.

Species		Amino Acid #
Mouse	**mgacriqyvl liflliasrw tlvqntycqv sqtlsleddp grtfnwtska**	50
Rat	**mgacriqyil lvflliashw tlvqniycev srtlslednp sgtfnwtska**	50
Human	**mgtpriqhll illvlgasll tsglelycqk glsmtveadp anmfnwttee**	50
Bovine	**mgachfqgll llflvgaptl imaqklfcqk gtfmgiqeda tnmfnwtsek**	50

Mouse	**-eqcnpgelcq etvllikadg trtvvlasks cvsqggeavt fiqytappgl**	100
Rat	**-ekcnpgefcq etvllikaeg tktailasks cvpqgaetmt fvqytappgl**	100
Human	**vetcdkgalcq etiliikag- tetailatkg cipegeeait ivqhssppgl**	100
Bovine	**veacdngtlcq etilliktag tktailatks csldgtpait fiqhtaapsl**	100

Mouse	**vaisysnycn dslcnnkdsl asvwrvpett a-tsnmsgtr- hcptcvalgsc- **	150
Rat	**vaisysnycn dslcnnrnnl asilqapept a-tsnmsgar- hcptclalepc-**	150
Human	**ivtsysnyce dsfcndkdsl sqfwefsett astvst--tl- hcptcvalgtcf**	150
Bovine	**aaisysnyce dpfcnnregl ydiwniqete eetkgt—tsl- hcptclalgsc**	150

Mouse	**ssapsmpcan gttqcyqgrl efsgggmdat vqvkgcttti gcrlmamids **	200
Rat	**ssapsmpcan gttqcyhgki elsgggmdsv vhvkgcttai gcrlmakmes**	200
Human	**sapslp-cpn gttrcyqgkl eitgggiess vevkgctami gcrlmsgila**	200
Bovine	**lnapsvacpn ntdrcyqgkl qvsegnvnsl leikgctsii gcklmsgvfk**	200

Mouse	**-vgpmtvketc syqsflqprk aeigasqmpt slwvlellfp- llllplth---fp**	250
Rat	**-vgpmtvketc syqsflhprm aeigaswmpt slwvlelllp- alslpliy---fp**	250
Human	**-vgpmfvreac phqlltqprk tengatclpi pvwglqlllp- ll-lpsfih--fp**	249
Bovine	**kigplwvketc psmsist-rk idngatwlht svwklklllm- llllilggsasgp**	253
**Species**		**Amino acid #**

Conserved cysteine residues are indicated by black boxes, and possible *N*-glycosylation sites (n-x-s/t) are dark-gray shadows.

**Table 3 ijms-21-06628-t003:** Possible proteins associated with TEX101 within the testis.

Protein Name	Antibody Used for the Experiments	Reference
Annexin A2	TES101	[11]
Ly6k	TES101	[11]
Cellubrevin	TES101	[11]
DPEP3	TES101, Ts4	[72]
5′-nucleotidase (CD73)	TES101, Ts4	[115]

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
