# Peer review of "Role of the Glycosylphosphatidylinositol-Anchored Protein TEX101 and Its Related Molecules in Spermatogenesis"

_ijms, 2020, doi:10.3390/ijms21186628_

Round 1
Reviewer 1 Report
In this reviewer article, Yoshitake and Araki describe the function and molecular mechanisms of TEX101 in spermatogenesis. The manuscript is well-written and provides detailed and important information for the field. Several points should be addressed before publication:
- Introduction: (1)The whole section should be reorganized. The authors should explain why we writing a review paper on a single protein is important. (2)Line 38: “ its ability to produce functionally intact gametes.” Tex101 is a protein, it cannot produce gamates. Please revise.
- Molecular distribution of TEX101 within gonadal organs: (1)“molecular distribution” is confusing, please revise. (2)Line 184 to 190 is hard to understand. For example, “was nearly identical among various stages, but varied among cell types in seminiferous tubules….”. Please write clearly what type of cells express TEX101 and if it is related with the stage of seminiferous epithelium.
- TEX101 function during fertilization: (1)should describe the detailed phonotype of Tex101 KO animal. Delete all the sentence about TEX101 because these information is in the previous section. (2)“TEX101 sugar chains” this paragraph is out of place. should move some information to “3.1 molecular ….”. and only discuss the function of sugar chains in fertilization.
Author Response
Responses to the Reviewer 1
Reviewer’s comments 1-A
In this reviewer article, Yoshitake and Araki describe the function and molecular mechanisms ofTEX101 in spermatogenesis. The manuscript is well-written and provides detailed and important information for the field. Several points should be addressed before publication:
Our responses 1-A:
We appreciate your encouragements for our manuscript. We have revised the manuscript according to your suggestions. See below in the details.
Reviewer’s comments 1-B:
- Introduction:
(1) The whole section should be reorganized. The authors should explain why we writing a review paper on a single protein is important.
Our responses 1-B:
Mammalian spermatozoa acquire fertilizing capacity as they cross the epididymis in vivo. However, spermatogenesis itself takes place in the testicular seminiferous tubules. Spermatogenesis, i.e., testicular mitotic proliferation, meiosis, sperm cell division, and morphological changes from haploid sperm to mature spermatozoa, results in the generation of cells that are highly specialized in structure and function. No other cell has undergone such extreme morphological changes while undergoing both genetic modification and reduced chromosomal pluripotency. It is generally believed that the testes have a unique mechanism to control spermatogenesis, where certain molecules are activated during germ cell production and others are repressed.
Concerning the research field, we have been studying the structure and function of testicular factors, especially the mechanism of germ cell production, starting with the discovery of a protein (TEX101, a glycosylphosphatidylinositol-anchored protein) in the past two decades. This protein is the core of our research, and we have accumulated a wealth of knowledge based on the functional relationship between TEX101, a GPI-AP and other molecules.
Therefore, we focus on TEX101 in this review, as a main target among testicular GPI-APs, and future avenues for investigation of testicular GPI-APs including potential role as regulators of ion channels are discussed.
We have rewritten the “Introduction” including above statements.
See Page 1, lines 31-38, and page 2, lines 44-47, in the revised manuscript.
Reviewer’s comments 1-C:
(2) Line 38: “its ability to produce functionally intact gametes.” Tex101 is a protein,
it cannot produce gametes. Please revise.
Our responses 1-C:
We agree with the concern. We would like to wording here to be "an important biomolecule for the production of normally functioning gametes". We have revised the text.
See page 2, lines 49-50, in the revised manuscript.
Reviewer’s comments 1-D:
- 2. Molecular distribution of TEX101 within gonadal organs:
(1) "molecular distribution" is confusing, please revise.
Our responses 1-D:
We have substituted "Molecular distribution" for "Subcellular localization" to be more specific.
See page 5, line 215, in the revised manuscript.
Reviewer’s comments 1-E:
(2) Line 184 to 190 is hard to understand. For example, “was nearly identical among various stages, but varied among cell types in seminiferous tubules….”.
Please write clearly what type of cells express TEX101 and if it is related with the stage of seminiferous epithelium.
Our responses 1-E:
The intensity of TEX101 immunofluorescence observed in the epithelium of seminiferous tubules is indistinguishable by weakly magnified microscopy regardless of the stage of the testicular epithelium. However, on a cellular level, we can clearly identify them.
We have stated these facts and details are available in the cited literatures.
See from page 5, line 230 to page 6, line 240, in the revised manuscript, and Refs [5,9,13,95].
(see also Reviewer 2, Comments D and our responses to the comments)
Reviewer’s comments 1-F:
- TEX101 function during fertilization:
(1) should describe the detailed phonotype of Tex101 KO animal. Delete all the sentence about TEX101 because these information is in the previous section.
Our responses 1-F:
According to the reviewer, we have re-written the text the detail phenotype of Tex101 KO animal, and all sentence concerning TEX101 have been deleted in this section.
See page 6, lines 271-277, in the revised manuscript (New Chapter 3.5).
Reviewer’s comments 1-G:
(2) "TEX101 sugar chains" this paragraph is out of place. should move some information to “3.1 molecular ….”. and only discuss the function of sugar chains in fertilization. 
Our responses 1-G:
We agree with the suggestion made by the reviewer. Therefore, we have moved this whole paragraph into Chapter 3.2"Molecular characteristics of TEX101, a unique glycoprotein germ cell marker". In addition, some information concerning functions of the bisecting GlcNAc have been added (as refs [99-101]).
See from page 4, line 184 to page 5, line 213, in the revised manuscript.

Reviewer 2 Report
Author performed lots of studies of TEX101 in reproductive system previously.
The manuscripts is well written, but additional information should be addressed to the manuscripts
1) there is not sufficient informaton for the function of TEX101 with its related molecules during gonad development both male and female
2) Author focused on Ly6k protein as TEX101 related moleculs, additional information of TEX101 related molecules except Lyk6 need in this manuscripts
3) Author described that diversity of TEX101 expression during testis and gonad development
TEX101 were detactable in prospermatogonia (12dpc-6dpp) and leptotene (12-14dpp), Zygotene (14dpp), Pachytene (147dpp)
(Taskeshi Takayama et al., Biology of Reproduction, 2005)
Author needs to more describe a diversity of TEX101 exprssion in testis development. Probably, the type of TEX101 related molecules were different in developmental stage of TEX101 postive cells. Author need to clarify on this.
Author Response
Responses to the Reviewer 2
Reviewer’s comments 2A:
Author performed lots of studies of TEX101 in reproductive system previously.
The manuscripts is well written, but additional information should be addressed to the Manuscripts
Our responses 2A:
Thank you for your favorable comments on our manuscript. We have revised according to your suggestions. See below in the details.
Reviewer’s comments 2B:
1) there is not sufficient informaton for the function of TEX101 with its related molecules during gonad development both male and female
Our responses 2B:
According to the suggestion made by the reviewer, we have added some statements in to the "3.5. TEX101 function during fertilization"
See page 6, line 271-284, in the revised manuscript.
Reviewer’s comments 2C:
2) Author focused on Ly6k protein as TEX101 related moleculs, additional information of TEX101 related molecules except Lyk6 need in this manuscripts
Our responses 2C:
According to the reviewer’s suggestion, we have added the information concerning Annexin A2, Cellubrevin, DPEP3, and CD73 in the end of "3.4. Molecules associate with TEX101 in male germ organ"
See page 6, lines 247-269, in the revised manuscript.
Reviewer’s comments 2D:
3) Author described that diversity of TEX101 expression during testis and gonad development TEX101 were detactable in prospermatogonia (12dpc-6dpp) and leptotene (12-14dpp), Zygotene (14dpp), Pachytene (147dpp) (Taskeshi Takayama et al., Biology of Reproduction, 2005) 
Author needs to more describe a diversity of TEX101 exprssion in testis development. Probably, the type of TEX101 related molecules were different in developmental stage of TEX101 postive cells. Author need to clarify on this.
Our responses 2D:
The intensity of TEX101 immunofluorescence observed in the epithelium of seminiferous tubules is indistinguishable by weakly magnified microscopy regardless of the stage of the testicular epithelium. However, on a cellular level, we can clearly identify them.
We have stated these facts and details are available in the cited literatures.
See from page 5, line 230 to page 6, line 240, in the revised manuscript, and Refs [5,9,13,95].
(see also Reviewer 1 comments E and our responses to the comments)
